# LARGE LANGUAGE MODELS AS GAMING AGENTS

## ABSTRACT

Although recent works have demonstrated that Large Language Models (LLMs) are starting to excel at following human instructions, their strategic thinking, planning, and long-term decision-making skills remain unclear. To rigorously evaluate these capabilities, we propose leveraging strategic gaming environments as they provide well-defined, structured benchmarks with clear success criteria. Specifically, we adopt multiple popular reasoning-oriented autonomous agents and analyze their performances over two popular strategic gaming environments: *Tic-Tac-Toe* (one of the most popular complete information gaming) and *Texas Hold'em Poker* (one of the most popular incomplete information games). To our surprise, we find that even one of the most advanced LLMs, ChatGPT, is largely ineffective in these two gaming scenarios. An even more surprising finding is that state-of-the-art reasoning methods, e.g., Chain-of-Thought, ReAct, etc., do not help much. For instance, in the naive 3×3 Tic-Tac-Toe environment, nearly all agents only perform slightly better than the random agent, i.e., the agent that randomly selects an action at each step. To understand this failure mode in more depth, we carry out a detailed demographic analysis. Our analysis uncovers two potential reasons behind this weakness: 1) autonomous agents lack gaming intents, i.e., they cannot "think ahead" to defend against opponents' future moves; 2) LLMs suffer from severe hallucinations and factual errors, e.g., even advanced reasoning agents fail to recognize immediate win/lose situations. With these insights, we take the first step to propose a simple yet effective **T**hink **A**head **La**nguage-powered **G**aming **A**gent (`TALAGA`). `TALAGA` recursively thinks ahead of the opponent's move, evaluates current gaming situations, and adjusts action selection with reward signal backtrack. We further empower `TALAGA` with additional features to alleviate hallucinations and factual errors, such as uncertainty estimation. Experimental results demonstrate that `TALAGA` significantly outperforms existing reasoning autonomous agents. A broader implication of our exploration is that games can serve as stress tests for LLMs, pushing them to their limits and uncovering vulnerabilities or weaknesses. We hope that this paper sheds new light on the limitations of current autonomous reasoning agents, which, in turn, would help with model improvements and achieve greater robustness.

## 1 INTRODUCTION

In recent years, Large Language Models (LLMs) have witnessed remarkable advancements in natural language inference and generation. Models such as ChatGPT have astounded us with their proficiency in unraveling the intricate nuances of human language. However, beneath their impressive linguistic capabilities lies a critical question that has piqued the curiosity of researchers and practitioners alike: "*what lies beyond language comprehension for LLMs?*"

While LLM's language-related feats have been well-documented and celebrated, their ability for higher-level reasoning and strategic planning remains relatively uncharted territory. Some recent works have devised approaches to improve the reasoning and planning capabilities of LLMs. For instance, Chain-of-Thought (`CoT`) Wei et al. (2022), Tree-of-Thought (`ToT`) Yao et al. (2023), and Graph-of-Thought (`GoT`) Besta et al. (2023) propose to reason in *step-by-step*-like manners. On the other hand, methods such as `ReAct` Yao et al. (2022) and `Reflexion` Shinn et al. (2023) prompt LLMs to "*reason before acting*" and "*self-reflection*" to accumulate insights and experiences that are used to improve behavior for the next trial. These reasoning approaches are shown to improve

performance on a range of benchmarks such as math word problems, symbolic reasoning, and even solitary games like Game of 24 and Mini Crosswords. However, as we steer the course of AI development towards more comprehensive assessments of cognitive prowess Sumers et al. (2023), it becomes increasingly apparent that these benchmarks, while valuable, possess inherent limitations. They often evaluate LLMs in isolation, without considering the dynamic and strategic interactions that frequently occur in real-world scenarios.

This paper responds to the imperative need for a more holistic evaluation framework by introducing a novel gaming benchmark that incorporates strategic opponents. Specifically, we propose to use two popular strategic gaming environments—*Tic-Tac-Toe* and *Texas Hold'em Poker*—as they provide well-defined, structured benchmarks with clear success criteria. In this context, the agent generates well-formatted *gaming path* during the gaming process. In this way, we are able to analyze LLMs' selection and probe their errors. Furthermore, our benchmarks also allow to use diverse opponents that could expose intricate hallucinations Ji et al. (2023) in LLM reasoning.

Through these benchmarks, we endeavor to provide a more nuanced understanding of LLM reasoning capabilities, addressing the pressing question of whether these models can navigate complex, interactive situations and outwit strategic opponents–an essential aspect of their potential applications in various real-world domains. To achieve this, we carry out a detailed performance analysis using one advanced LLMs, such as ChatGPT , and evaluate the performance of several state-of-the-art reasoning schemes against diverse opponents.

To our surprise, we found that most of these reasoning methods are largely ineffective in our strategic gaming scenarios. For instance, in the naive $3 \times 3$ Tic-Tac-Toe environment, nearly all state-of-the-art reasoning methods only perform slightly better than the random opponent, i.e., the agent that randomly selects an action at each step. With detailed demographic analysis, we conclude the main reasons behind this failure model as two-fold: ❶ *Autonomous agents lack gaming intent*, i.e., they cannot *think ahead* to defend the future moves from their opponents; ❷ *LLMs suffer from severe hallucinations Duan et al. (2023); Manakul et al. (2023) and factual errors Bian et al. (2023); Karpinska & Iyyer (2023); Gekhman et al. (2023)*, e.g., LLMs cannot recognize immediate win situations (whether two/three symbols are in a row for Tic-Tac-Toe), which makes them lose even with reasonable planning or thoughts.

To overcome these limitations, we propose a simple yet effective **T**hink **A**head **La**nguage-powered **G**aming **A**gent (`TALAGA`). `TALAGA` uses a recursive prompting mechanism that automatically analyzes the opponents' potential future moves/actions and assigns reward signals for these situations. Then, the reward signal is backtracked to the current action and eventually dictates the action selection of `TALAGA`. To alleviate hallucinations and factual errors, we integrate uncertainty estimation in our framework so that `TALAGA` can carry out calibrated reasoning. Under comprehensive gaming settings, `TALAGA` significantly outperforms state-of-the-art reasoning approaches. Our key contributions are summarized as follows:

- **Strategic Gaming Benchmarks:** We propose to use strategic gaming benchmarks for a comprehensive assessment of reasoning capabilities of LLMs.
- **Unraveling Failure Mode of LLM Reasoning:** We found that most of the state-of-the-art reasoning methods are largely ineffective in our strategic gaming scenarios.
- **Insights Into the Failure Mode:** We identified the reasons behind this failure model – unable to think ahead and severe hallucinations.
- **An Improved Reasoning Strategy:** To fix these issues, we propose a novel reasoning method referred to as `TALAGA`. Experimental results demonstrate that our method significantly outperforms existing reasoning approaches.

## 2 RELATED WORKS

**Reasoning and Planning with LLMs.** Recently, LLMs have demonstrated reasoning and planning abilities by breaking down intricate questions into sequential intermediate steps, known as Chain-of-Thought (`CoT`) Wei et al. (2022), prior to generating the final response. Building upon this concept, Self-Consistency Wang et al. (2022) samples multiple chains and selects the best answer via majority voting, `ToT` Yao et al. (2023) models the LLM reasoning process as a tree structure, GoT Besta et al.

| Agent v.s. Agent | Random | MinMax | Prompt | CoT | CoT-SC | ToT | ReAct | Avg. Win Rate |
|---|---|---|---|---|---|---|---|---|
| Random | - | 4.50% | 40.00% | 36.50% | 37.50% | 33.50% | 37.50% | 31.58% |
| MinMax | 86.00% | - | 92.00% | 83.50% | 85.00% | 81.50% | 76.00% | 84.00% |
| Prompt | 54.50% | 5.00% | - | 24.00% | 20.00% | 24.00% | 24.50% | 25.33% |
| CoT | 54.00% | 4.50% | 43.50% | - | 36.00% | 42.50% | 39.00% | 36.58% |
| CoT-SC | 52.50% | 7.00% | 38.00% | 36.00% | - | 31.50% | 36.00% | 33.50% |
| ToT | 55.00% | 8.00% | 52.00% | 30.00% | 29.00% | - | 48.00% | **37.00**% |
| ReAct | 54.00% | 6.00% | 38.50% | 39.00% | 33.50% | 38.50% | - | 34.92% |
| **Avg. Loss Rate** | 59.33% | 5.83% | 50.67% | 41.50% | **40.17**% | 41.92% | 43.50% | - |

Table 1: Benchmark existing reasoning agents over the Tic-Tac-Toe environment. Note that the game result can be draw, so the sum of the Win Rate of a pair of two agents is not 100%. It is shown that existing autonomous reasoning agents are largely ineffective for the simple 3×3 Tic-Tac-Toe gaming task. Only `ToT` and `CoT` outperform `Random` agent with moderate margins and all other agents are just slightly better or even worse than `Random`.

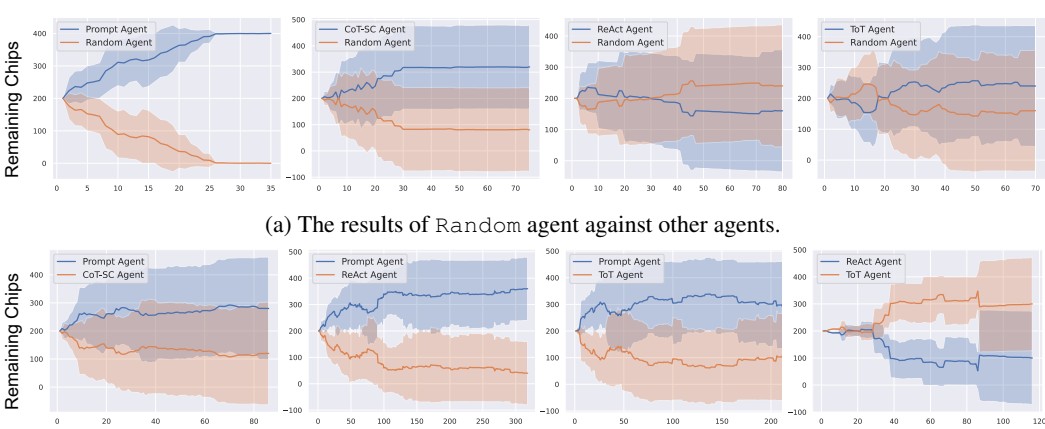

(a) The results of `Random` agent against other agents.

(b) The results of `Prompt` agents against other agents and `ReAct` v.s. `ToT` in the last column.

Figure 1: **X-axis:** the number of hands, **Y-axis:** the remaining chips for each player. The performance of reasoning agents in the Texas Holdem Poker environment. The performance of autonomous agents in Texas Holdem Poker gaming. Standard derivations over 10 trials are shown in shadow areas. Among these agents, `Prompt` works better than other methods and `ReAct` is slightly worse than `Random`.

(2023) models the information generated by an LLM as an arbitrary graph, where the information elements as vertices, and dependencies as edges. In an orthogonal direction, RoT Lee & Kim (2023) encourages LLMs to segment a problem into multiple contexts, enabling context-specific operations. Kojima et al. (2022) has demonstrated the proficiency of LLMs as zero-shot reasoners by simply adding "Let's think step by step" before each answer. In addition, LLMs have achieved successful results in planning and action generation Wu et al. (2023); Huang et al. (2022). Driess et al. (2023) proposes a multimodal language model for embodied reasoning tasks, visual-language tasks, and language tasks. Beyond that, Liu et al. (2023a) translates such intermediate steps into executable programming languages to conduct classical planning algorithms.

**LLMs as Autonomous Agents.** Conventional reinforcement learning based frameworks have exhibited great achievement in Autonomous Agents Silver et al. (2016); Schrittwieser et al. (2020); Zhao et al. (2022), but require considerable training cost. Recently, Autonomous Agents have driven zero/few-shot LLMs to achieve complex reasoning and planning tasks through prompt engineering Liu et al. (2023b); Xi et al. (2023); Xiang et al. (2023). Yao et al. (2022); Shinn et al. (2023) endow agents with the capability to engage in introspection regarding the feedback provided by LLMs. Zhou et al. (2023) completes tasks with an unprecedented degree of autonomy. Cai et al.

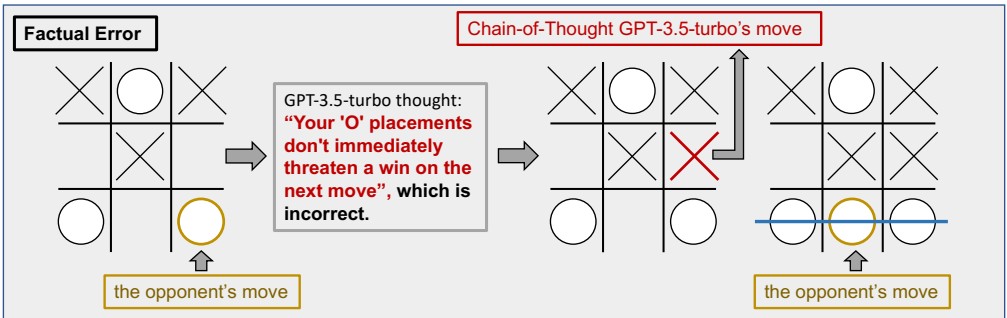

(a) *factual error*: LLMs failed to recognize immediate win/lose situations.

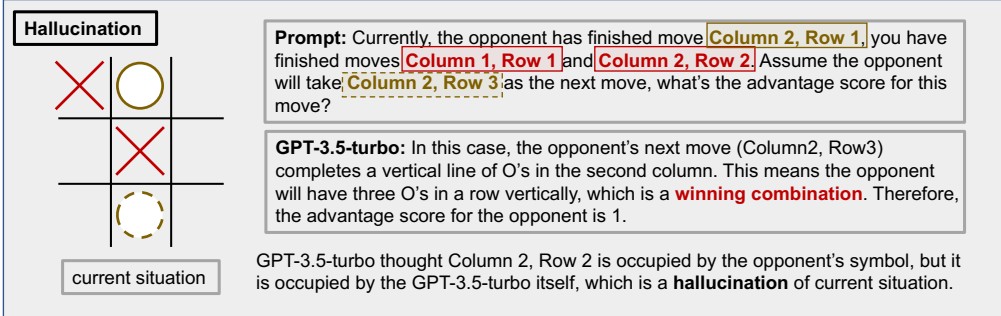

(b) *hallucination*: LLMs mistake his own chess/symbol for his opponent's.

Figure 2: Some representative failure patterns of GPT-3.5-turbo in a $3 \times 3$ Tic-Tac-Toe game.

(2023) proposes a closed-loop framework empowering large language models (LLMs) to create and utilize their own tools for diverse tasks. Lin et al. (2023); Feng et al. (2023) leverage human interaction with LLMs. Furthermore, multi-agent systems are also empowered by LLMs, Shen et al. (2023) leverages ChatGPT to connect various AI models in Hugging Face to solve AI tasks. Within the studies in Li et al. (2023); Park et al. (2023), cooperative behaviors emerge autonomously within a group of agents, exemplifying the spontaneous development of teamwork dynamics. Chen et al. (2023) can leverage group cooperation to handle more complex scenarios and dynamically adjust its composition according to the current state. Although existing LLM-based autonomous agents have demonstrated remarkable success in addressing complex tasks, their applicability to dynamic gaming scenarios remains limited, thereby motivating the development of our `TALAGA` framework.

## 3 BENCHMARK SOTA AUTONOMOUS REASONING AGENTS

In this section, we will introduce our benchmark specifications and conduct comprehensive experiments to analyze the performance of existing reasoning agents over rigorous gaming scenarios (Tic-Tac-Toe and Texas Hold'em Poker[1]).

### 3.1 PRELIMINARY

There are two rigorous gaming tasks in our environment: Tic-Tac-Toc and Texas Hold'em Poker. For Tic-Tac-Toe, we will utilize the version of $3 \times 3$ grid with the winning length as 3. The first-go player will obtain large advantages in this game: 1) he will have one more chance to place the symbol; 2) he can choose to take the strategic center position. Therefore, when performing agent-to-agent experiments, we will execute 200 matches with each agent going first for 100 matches. Then, we use the average rates of win and loss as the metrics to evaluate performance.

In terms of Texas Hold'em Poker, for simplicity, we only adopt 8 actions in the game, including `FOLD`, `CHECK`, `CALL`, `RAISE_3BB`, `RAISE_HALF_POT`, `RAISE_POT`, `RAISE_2POT`, `ALL_IN`, `SMALL_BLIND`, `BIG_BLIND`. Detailed explanations of these actions can be found in Appendix B.

---

[1]`https://github.com/dickreuter/neuron_poker`

Initially, each agent will be assigned $200 chips. As the game goes on, the agent can either earn chips from others or lose their chips, and the game finishes when only one player has chips. Therefore we will utilize the *number of being final winner* as the main metric.

Besides, we mainly consider 7 agents (5 LLMs-powered agents and 2 baseline agents) and investigate their performance over our rigorous gaming environments: ❶ `Random`: the agent that randomly selects action at each step; ❷ `MinMax`: the agent that selects action based on conventional optimization-based minimax gaming strategy (only compatible with Tic-Tac-Toe); ❸ `Prompt`: the agent that is directly asked to return answers, without planning or thoughts; ❹ Chain-of-Thought (`CoT`): the agent that reasons through thinking step by step; ❺ Self-Consistent CoT (`CoT-SC`): the agent that utilizes multiple step-by-step-thinking trajectories (the number of trajectories is set to 3 by default) during reasoning; ❻ Tree-of-Thought (`ToT`): the agent that augmented with exploration and deliberate decision-making, i.e., self-evaluation (the number of generations and votes are set to 3 by default). ❼ `ReAct`: the agent that follows reasoning-before-acting policy. We use **gpt-3.5-turbo** to drive all the agents.

It is worth noting that some agents are not originally designed for interactive gaming tasks. For instance, `ReAct` requires a pre-defined action space for `acting`. In Appendix C, we provide how we make them available to our gaming environment for each method in detail.

## 3.2 EMPIRICAL EVALUATION

We first conduct agent-to-agent Tic-Tac-Toc matches and report their average win rate and average loss rate. Results are summarized in Table 1. Surprisingly, most advanced reasoning agents work only slightly better than `Random` agent. `Prompt` works even worse than `Random`. Among these methods, `ToT` achieves the highest average win rate (37%) and `CoT-SC` achieves the lowest loss rate (40.17%). Based on these results, we can conclude that even though 3×3 Tic-Tac-Toe has very limited search space and action space, it still remains challenging for LLMs-powered reasoning agents.

In Figure 1b, we present the performance of reasoning agents when playing Texas Hold'em Poker in an agent-to-agent manner. We provide the performances of `Random` v.s. other agents and `Prompt` v.s. other agents. We found that `Prompt` works better than others, even for those with advanced reasoning skills. `ReAct` is slightly worse than `Random` agent.

## 3.3 ANALYTICAL INSIGHTS

According to the above results, here we provide possible reasons why existing agents are not suitable for our rigorous gaming scenarios. With careful human examinations, we summarize the following potential reasons for the limited success of reasoning agents over rigorous gaming tasks:

**Lacking Gaming Intent.** We found that existing reasoning agents can only reason based on the current state, rather than think ahead to defend against the opponent's future moves. We believe it is because reasoning agents are designed for complex planning or reasoning tasks, such as Game of 24 Yao et al. (2023), ALFWorld Shridhar et al. (2020), where there is no opponent in these situations, rather than gaming.

**Hallucination and Factual Error.** We found that a lot of hallucinations and factual errors occur during reasoning, such as the provided representative examples in Figure 2b.

Apart from these limitations, we also found that reasoning is important for gaming tasks, e.g., methods like `CoT`, `CoT-SC`, `ToT` all work better than the naive `Prompt`. However, advanced reasoning methods do not always work better, e.g., `CoT-SC` is worse than `CoT` in both Tic-Tac-Toe and Texas Hold'em Poker. One of the possible reasons is that multi-step reasoning is more likely to introduce more hallucinations, which eventually hurt the performance.

## 4 TALAGA: THINK AHEAD LANGUAGE-POWERED GAMING AGENT

As mentioned in Section 3.3, existing reasoning agents have limited effectiveness for rigorous gaming tasks suffering from weak gaming intent and serious hallucinations and factual errors. In this

section, we introduce our TALAGA in detail. Specifically, we first formally describe the language-powered gaming process. Then we introduce our algorithm of recursively thinking ahead and also advanced techniques to mitigate hallucinations and factual errors. The structure of TALAGA is presented in Figure 3, to provide an intuitive understanding.

## 4.1 FORMULATION OF LLM-POWERED GAMING PROCESS

We formulate the gaming process as a discrete decision-making process among *actors*, under the interaction with the gaming environment. We define *actors* as LLMs-powered agents that take natural language (or prompts) as inputs and generate corresponding actions as outputs. Without loss of generality, we assume there are two actors participating in this gaming process. Considering the $t$-th step of this process, we denote by $s_t \in \mathcal{S}$ the state that the two actors observed, and $a_t, \hat{a}_t \in \mathcal{A}$ the actions sampled by the two actors, where $\mathcal{S}$ is the infinite state space and $\mathcal{A}$ is the infinite action space. The state transition kernel from $s_t$ to $s_{t+1}$ can be formulated as $s_{t+1} = \mathcal{T}(s_t, a_t)$ where $a_t \sim p(a_t|s_t, x)$, $p(a_t)$ refers to the generative distribution of the backbone LLMs, and $x$ is the necessary instructions, e.g., few-shot prompts, chain-of-thought prompts. In this way, the two-agents gaming process can be represented as the sequence $(s_0, a_0, s_1, \hat{a}_1, s_2, \cdots, s_N)$, where $s_0$ is the initial state and $s_N$ is one of the terminal states, e.g., a win/draw/loss situation. In this process, the two actors will alternatively sample actions to achieve new states, aiming to maximize their winning rates.

We denote external instructions by $x$, which will be fed into actors to achieve various purposes, e.g., few-shot prompts, chain-of-thoughts, and instruction following. For simplicity, we do not distinguish these instructions in this section and omit the explanation of $x$ unless it is necessary. We provide all the $x$ utilized in this section in Appendix A.

## 4.2 RECURSIVELY THINK AHEAD

Foresight is one of the significant differentiating factors between top-tier players and average players, especially in strategy games like chess and card games. It requires the players to calculate moves ahead, visualize the board's possible states, and evaluate the consequences of various move sequences. To simulate this process, we formulate TALAGA as the ensemble of modules, utilizing multiple individual actors:

- **Main Actor** $M$: interacting with the environment, gathering feedback from other actors, and generating the next action, i.e., $a_t \sim P_M(a_t|s_t, x)$ where $s_t$ is the current state and $x$ is the external instructions/feedback.

- **Reward Actor** $M_R$: working as a signal function to evaluate the reward signals of different actions, i.e., $r \sim P_R(r|s_t, x)$.

- **Anticipation Actor** $M_O$: an imaginary opponent, predicting action $a_{o,t}$ to beat $M$ at state $s_t$, i.e., $\hat{a}_{o,t} \sim P_O(\hat{a}_{o,t}|s_t, x)$.

Here $P_M, P_R$ and $P_O$ are the generative distributions of the backbone LLMs for $M, M_R$ and $M_O$, respectively.

Some conventional optimization based multi-agent gaming frameworks, such as minimax gaming Lan et al. (2020), are standard think-ahead frameworks. In these frameworks, both actors, assumptive to be sufficiently smart, will always sample the actions that maximize their own rewards. Our recursively-think-ahead mechanism follows this simple and classic protocol. Specifically, at the beginning of each gaming step $t$, we first sample $n$ desired actions $\mathcal{A}_t = \{\tilde{a}_t^1, \tilde{a}_t^2, \cdots, \tilde{a}_t^n\} \sim P_M(\tilde{a}_t|s_t, x)$ from $M$ as the candidacy actions, given current state $s_t$. Then, we formulate the think-ahead process as the *pseudo-gaming* with $M_O$, as the following sequence:

$$(s_t, \tilde{a}_t, s_{t+1}, \hat{a}_{o,t+1}, s_{t+2}, \tilde{a}_{t+2}, \cdots, s_T), \tag{1}$$

where $\tilde{a}_t \in \mathcal{A}_t$ is a candidacy action at pseudo-gaming step $t$, $\hat{a}_{o,t+1} \sim P_O(\hat{a}_{o,t}|s_{t+1}, x)$ is the sampled action from imaginary opponent $M_O$, and $s_T$ is a terminal state, e.g. achieves win/draw/lose situation or achieves state $s_{t+k}$ where $k$ is the maximum allowed number of think-ahead steps. Once the terminal state is achieved in pseudo-gaming, the reward agent $M_R$ will perform situation assessment by answering an advantage score, $r_T$, to describe how many advantages the actor $M$

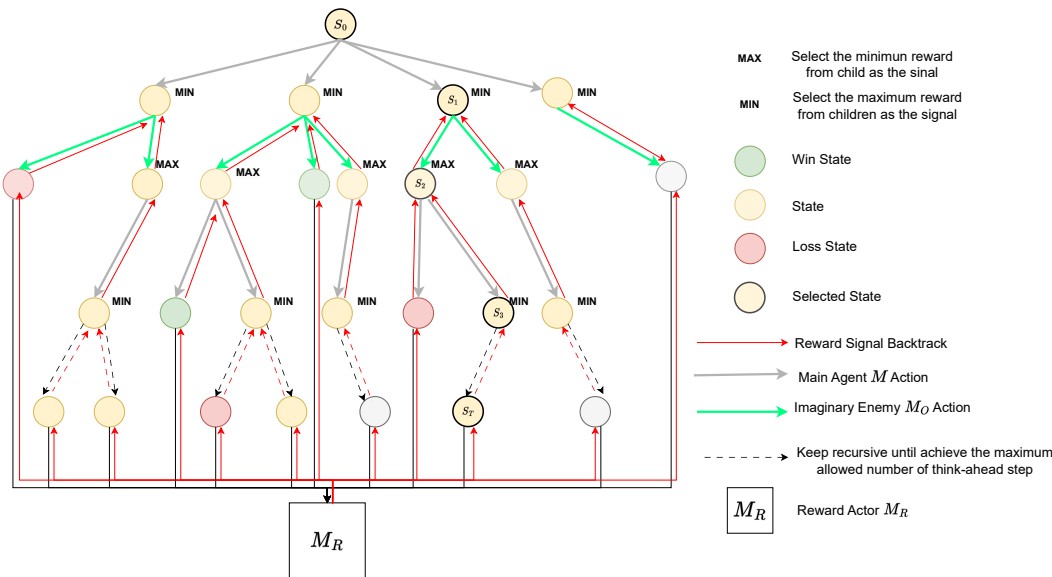

Figure 3: The tree representation of TALAGA. Each trajectory is equivalent to a pseudo-gaming sequence, as described in Eq. (1). TALAGA reasons in a recursive manner.

has at state $s_T$: $r_{s_T} \sim P_o(r_T|s_T, T, x)$, $(0 \leq r \leq 1)$. Theoretically, if we traverse all the possible combinations of candidacy actions and always take $k$ steps to achieve terminal states, there will be a $k$-layer decision-making tree constructed with $n^k$ leave nodes, which indicates there will be at most $n^k$ terminal states and advantage scores in total.

Once we finish traversing this decision tree and obtain advantage scores for each terminal state, we will perform reward signal backtrack from state $s_T$ to $s_t$ and select action $a_t$, in a minimax manner:

$$\max_{a_t \in \mathcal{A}} \min_{\hat{a}_{t+1} \in \mathcal{A}} (r_{s_t} P_O(\hat{a}_{t+1}|s_{t+1}) P_M(a_t|s_t)). \tag{2}$$

With this minimax reward backtrack, we assume that the opponent will always choose the "worst case" during the gaming, which makes our agent more robust to the opponents. Once the traceback happens to the root of the tree, there will be a reward signal for each candidacy action in $\mathcal{A}_t$. Then, we simply select the action with the highest rewards as the next action. The key design of recursively thinking ahead is the imaginary opponent $M_O$ that tries to block the winning of $M$ and the minimax reward signal backtracking.

### 4.3 MITIGATING HALLUCINATION AND FACTUAL ERRORS

As we mentioned in Section 3.3, LLMs suffer from serious hallucinations and factual errors. Even in the simplest $3 \times 3$ Tic-Tac-Toe situation, LLMs struggle to read the correct spatial information and recognize/judge the immediate win positions. To mitigate this issue, we adopt two strategies during out gaming process: *majority vote* Wang et al. (2022) and *perturbation-based uncertainty estimation* Manakul et al. (2023).

For the majority vote, we simply sample multiple generations as options and let LLMs select the high-frequency option or the mean value if it is a numerical situation. For perturbation-based uncertainty estimation, we first prompt LLMs to perturb the target questions while keeping the semantics unchanged, then we sample generations based on both original question and perturbed questions and apply a majority vote over these generations.

It is worth noting that both majority vote and uncertainty estimation are general methods that can be applied to any prompting process. In our implementation, we only apply the two methods over the situation assessment procedures, i.e., generating advantage scores with reward actor $M_R$.

| Agent-to-Agent Pair | Avg. Win Rate of TALAGA | Avg. Win Rate of Others | Avg. Draw Rate |
|---|---|---|---|
| TALAGA v.s. ToT | **56%** (+21%) | 35% | 9% |
| TALAGA v.s. CoT-SC | **52%** (+17%) | 35% | 13% |
| TALAGA v.s. ReAct | **50%** (+13%) | 37% | 13% |
| TALAGA v.s. Prompt | **60%** (+26%) | 34% | 6% |
| TALAGA v.s. CoT | **59%** (+29%) | 30% | 11% |

Table 2: The performances of TALAGA when taking the agents as the opponent, over 100 Tic-Tac-Toe matches. It is shown that the proposed method significantly outperforms existing agents in rigorous gaming scenarios.

| TALAGA v.s. ToT | TALAGA Avg. WR | ToT Avg. WR | TALAGA Go First | | ToT Go First | |
|---|---|---|---|---|---|---|
| | | | TALAGAWM | ToTWM | ToTWM | TALAGAWM |
| TALAGA ($k = 2, n = 2$) | 37% | 59% | 30 | 17 | 42 | 7 |
| TALAGA ($k = 2, n = (4/2)$) | 48% | 37% | 35 | 8 | 29 | 13 |
| + hallucination control | 56% | 35% | 39 | 9 | 26 | 17 |

Table 3: The ablation study of TALAGA, over the Tic-Tac-Toe environments, when against ToT. **WR** stands for win rate and **WM** stands for the number of winning matches. $n =(4/2)$ means only sample 4 candidacy actions for the first time and the following candidacy actions are 2, as shown in Section 5.1.

## 5 EMPIRICAL RESULTS

In this section, we empirically evaluate the performance of TALAGA. We will first introduce the experimental settings. Then the evaluation in Tic-Tac-Toe and Texas Holdem will be presented separately. We also conduct detailed ablation studies to factorize the designing of TALAGA and individually evaluate their effectiveness.

### 5.1 EXPERIMENTAL SETTINGS.

**Running Trials.** For the agent-to-agent experiments, we will perform 100 matches per pair for Tic-Tac-Toe. To mitigate the first-go advantages, each agent will be the first to go in 50 matches. In terms of Tic-Tac-Toe experiments, we perform 10 trials for each agent-to-agent pair. For each trial, there are dozens of hands during the gaming.

**Hyperparameters.** There are two main hyperparameters introduced in the think ahead design of TALAGA: the maximum allowed steps to think ahead $k$, and the number of candidacy actions $n$. For all the Tie-Tac-Toe experiments, we set $k$ to be 2, i.e., at most think ahead 2 steps and we set $n$ to be 4 for the first candidacy action sampling and reduce it to half for all the following sampling. We will use $n = (4/2)$ to represent this configuration. In terms of majority vote and perturbation-based uncertainty estimation, we set the number of votes as 3 and only perturbed question once with 2 new generations regarding the perturbed question. All the agents are driven by gpt-3.5-turbo, with the temperature as 0.2.

### 5.2 TIC-TAC-TOE

We conduct experiments by taking TALAGA against ToT, CoT-SC, ReAct, Prompt, CoT. Results are summarized in Table 2. It is shown that our TALAGA significantly outperforms existing reasoning agents. For instance, TALAGA outperforms the most powerful agent ToT (as ToT achieves the best performance in Table 1) by 21%, when compared with win rate.

### 5.3 ABLATION STUDY

We individually investigate the effectiveness of each module in TALAGA. Apart from the average win rate, we also consider how many matches are won by the first-go agent. Results are presented in Table 3. For the average win rate, we show that increasing the number of candidacy actions will substantially increase the average win rate of TALAGA, which indicates the effectiveness of our

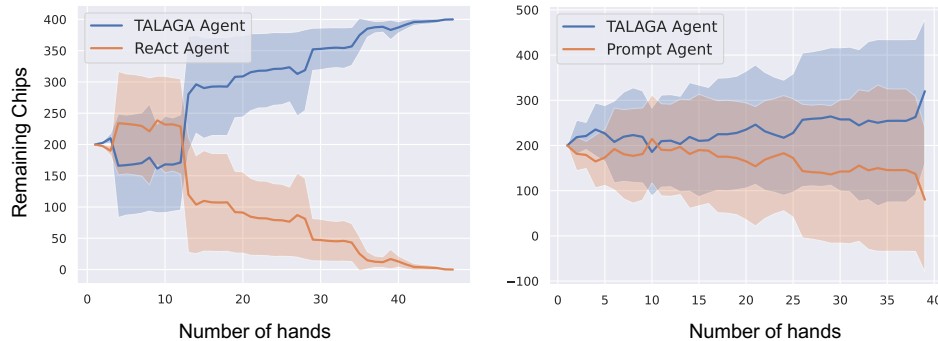

Figure 4: The performances of `TALAGA` v.s `Prompt` and `ReAct` in Texas Hold'em Poker.

| Agent-to-Agent Pair | **Win Rate of** `TALAGA` | **Win Rate of** others |
|---|---|---|
| `TALAGA` v.s. `Prompt` | 53.8% | 46.2% |
| `TALAGA` v.s. `CoT-SC` | 63.2% | 36.8% |
| `TALAGA` v.s. `ToT` | 72.1% | 27.9% |
| `TALAGA` v.s. `ReAct` | 78.0% | 22.0% |

Table 4: The hands win rate of `TALAGA` v.s. other baseline agents in Texas Hold'em Poker.

design. In addition, the average win rate will be further boosted by incorporating our hallucination reduction strategy.

### 5.4 TEXAS HOLD'EM POKER

Here we present the results under the evaluation of Texas Hold'em Poker. In Table 1, we show that `TALAGA` significantly outperforms existing reasoning agents. We also report the win rate of hands, by counting the total hands among all the trials and how many hands are won by `TALAGA`. Results are presented in 4. It is shown that `TALAGA` consistently outperforms existing reasoning methods in the hands win rate.

## 6 CONCLUSION

In this paper, we take the initial exploration of the potential utility of zero-shot LLMs as gaming agents. Our investigation involves the assessment of the performance exhibited by state-of-the-art reasoning and planning agents when applied to two gaming scenarios, Tic-Tac-Toe and Texas Hold'em Poker. Surprisingly, our findings reveal that most of these agents exhibit only marginally better than random agents. Notably, these existing agents manifest limitations in their capacity to "think ahead", and suffer from severe hallucinations and factual errors. Motivated by these observations, we introduce the Think Ahead Language-powered Gaming Agent (`TALAGA`), which represents a pioneering endeavor in harnessing language-powered models for the gaming context. The proposed `TALAGA` outperforms `ToT` by 21%, `CoT-SC` by 17%, and `CoT` by 29%, under the evaluation of Tic-Tac-Toe gaming. Empirical results derived from our experiments serve to substantiate the efficacy of our proposed approach.

**Limitations and Ethics Statement.** In this pioneering study focused on LLMs in gaming contexts, our examination is only centered on Chat-GPT, a highly representative LLM. Nevertheless, the performance evaluation of other LLMs such as LLaMA and Vicuna remains uncertain. Additionally, our investigation encompasses two gaming scenarios: Tic-Tac-Toe and Texas Hold'em Poker, chosen to represent complete and incomplete information systems, respectively. It is worth noting that numerous other gaming conditions merit exploration. It is imperative to emphasize that our research is conducted with a strong commitment to ethical principles. We do not intend to design any aspects of our experiments that could enable unfair play. Nonetheless, it is important to acknowledge that further developments in this field may raise concerns regarding potential harm, offensive, or unethical conduct within future gaming environments.

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

APPENDIX

## A  PROMPTS

Here we provide all the prompts we designed and utilized in this paper:

```
System Prompt:
You are a helpful assistant that strictly follows the
user's instructions.

Head Prompt:
Tic Tac Toe is a two-player game played on a grid.
Players take turns marking a space with their respective
symbol. The goal is to get multiple of one's own symbols
in a row, either horizontally, vertically, or diagonally,
before the opponent does. If all nine squares are filled
and no player has three in a row, the game is a draw.
The Tic Tac Toe game is played on a {grid_size} by {grid_size}
grid, with the winning length as {win_len}. Each move is represented
by a string consisting of two parts: the column (C) and the row (R),
in that order. For instance, C1R2 means the movement at the position
of the first column and the second row of the grid. You are
playing this game with the user (opponent).

Direct Prompt (first go):
You are the first to go. The legal positions are {legal_positions}.
Choose one move from these legal positions to set up advantages.

Your output should be of the following format:

Move:
Your move wrapped with <>, e.g., <C1R1>, <C1R2>, <C1R3>

(non first go)
Now, your opponent has finished moves: {opponent_moves}.
You have finished moves: {agent_moves}. The legal positions are
{legal_positions}. Choose one move from these legal positions to
set up advantages.

Your output should be of the following format:

Move:
Your move wrapped with <>, e.g., <C1R1>, <C1R2>, <C1R3>

Chain-of-Thought:
Now, your opponent has finished moves: {opponent_moves}.
You have finished moves: {agent_moves}.

The legal positions are {legal_positions}. First think about your
current situation, then choose one move from legal positions
to set up advantages.

Your output should be of the following format:

Thought:
Your thought.

Move:
Your move wrapped with <>, e.g., <C1R1>, <C1R2>, <C1R3>
```

```
Situation Assessment:
Given this situation, analyze your winning rate and provide a
concrete value (from 0 to 100). Conclude in the last line
"The winning rate is {s}", where s the integer id of the choice.

Reward Assessment:
Assume {side} will take {next_move} as the next move.
What is the advantage score of {side} for this move?
Use a score on a scale of 0 - 100 to represent this score.
Conclude in the last line "The advantage score for {side}
is {{s}}", where s is the score.
```

## B  TEXAS HOLDEM POKER

The explanations of Texas Holdem Poker actions:

1. FOLD: You decide not to play the hand and discard your cards.
2. CHECK: Declining the opportunity to bet. It's like saying 'I'm still in the game, but I don't want to bet right now.
3. CALL: Matching the current highest bet to stay in the hand.
4. RAISE_3BB: Raising the bet to three times the big blind amount.
5. RAISE_HALF_POT: Raising to an amount equal to half the current pot size.
6. RAISE_POT: Raising to an amount equal to the current pot size.
7. RAISE_2POT: Raising to an amount equal to twice the current pot size.
8. ALL_IN: Betting all your chips.
9. SMALL_BLIND: A forced bet that's typically half the size of the big blind. It rotates around the table.
10. BIG_BLIND: A forced bet that sets the initial pot amount and action. It's typically twice the size of the small blind and rotates around the table.

## C  MAJOR ADAPTIONS TO AUTONOMOUS AGENTS

For `CoT`, `CoT-SC`, and `Prompt`, please refer to A for which prompts are utilized. It is because prompts are the most important part for these methods.

For `ReAct`, we follow the prompts from their official codebase and utilize the first-think-then-action procedures. However, one of the major challenges is that we need to design search spaces for our tasks. For example, in Yao et al. (2022), the action space defined for the Hotpot QA dataset is SEARCH[entity], LOOKUP[entity], and FINISH. To do that, we design the following actions for Tic-Tac-Toe:

```
(1)Defensive Action, which means to block the potential winning of
your opponent (e.g., block your opponent from forming sequences of 3).
(2)Offensive Action, which means to win the game (e.g., create forks,
control the center, play ahead).
```

In terms of Texas Hold'em poker, we reuse the action presented in Appendix B.

For `ToT`, we follow the implementation of the text generation task. Specifically, follow the 2-step `ToT` manner, i.e., 1) generate plans; 2) vote for the plan; 3) generate action according to plan; 4) vote for action. The prompts used in this process are shown as follows:

```
FIRST GO Prompt:
You are the first to go. The legal positions are {legal_positions}.
First think about your current situation,
then choose one move from legal positions to set up advantages.

Your output should be of the following format:

Thought:
Your thought.

Move:
My move is <CxRy> ('<' and '>' are mandatory), where x and y are
the index of column and row, respectively.

Vote Prompt:
Conclude in the last line "The best choice is {s}",
where s is the integer id of the choice.
```

