# OpenReview forum: "Large Language Models as Gaming Agents"
_ICLR.cc/2024/Conference — ICLR 2024 Conference Withdrawn Submission_

### Official Review · Reviewer_MMad · 2023-10-17

**Soundness:** 2 fair
**Presentation:** 3 good
**Contribution:** 2 fair
**Rating:** 6
**Confidence:** 4

**Summary:**

This paper proposes using strategic gaming environments like Tic-Tac-Toe and Texas Hold'em poker as benchmarks to evaluate the reasoning and planning capabilities of large language models (LLMs). The authors test several state-of-the-art reasoning techniques like Chain-of-Thought and Tree-of-Thought in these gaming settings against different opponents. Surprisingly, they find most agents perform only slightly better than random, unable to "think ahead" against opponents and suffering from hallucinations. To address this, they introduce the Think Ahead Language-powered Gaming Agent (TALAGA) which recursively anticipates opponent moves, evaluates reward signals, and selects optimal actions. Experiments show TALAGA outperforms existing reasoning methods in both Tic-Tac-Toe and poker scenarios.

**Strengths:**

1) Introduces new gaming benchmarks to assess strategic capabilities of LLMs beyond just language tasks.

2) Provides interesting analysis unveiling limitations of current reasoning methods like lack of gaming foresight and factual errors.

3) Proposes a new agent, TALAGA, that demonstrates good performance in competitive gaming settings compared to existing approaches.

**Weaknesses:**

1) Applies approaches like CoT and ToT to gaming though not originally designed for this, perhaps unfair comparison.

2) Experiments limited to just two small game environments; more diversity could strengthen conclusions.

**Questions:**

1)  I think ChatGPT's poor performance in these games is mostly due to the limited availability of datasets for these games on the internet. If data from these games were included in ChatGPT's training data, I believe it could significantly reduce hallucinations and errors.

2) For poker, using the number of wins and losses as an evaluation metric is not particularly appropriate. It's better to use the amount of chips won or lost.

3) *Apart from these limitations, we also found that reasoning is important for gaming tasks, e.g., methods like CoT, CoT-SC, ToT all work better than the naive Prompt.* This statement seems to be contradictory to the experimental results in Figure 1.

4) Formula 2 lacks detailed explanation.

5) In order to make the article self-contained, it is better to provide a detailed explanation in the appendix of how majority vote and perturbation-based uncertainty estimation are implemented.

6) As I mentioned in the weakness section, the comparative methods in the paper, such as ToT and CoT, were not specifically designed for gaming environments. Therefore, they may not effectively measure the true performance of TALAGA. I am interested in understanding how TALAGA performs in comparison to state-of-the-art methods, for instance, comparing it to the CFR algorithm used in poker.

---

### Official Review · Reviewer_Fp2o · 2023-11-01

**Soundness:** 2 fair
**Presentation:** 2 fair
**Contribution:** 2 fair
**Rating:** 3
**Confidence:** 4

**Summary:**

The authors test one LLM model on two simple reinforcement learning tasks and find that it doesn't perform very well. They then show that by doing a tree search the model can be slightly improved.

**Strengths:**

The authors present a new technique for prompting LLMs and test it on a small task. The new prompting method outperforms the others on the tested games.

**Weaknesses:**

The paper presents the results entirely through the framing of language modelling, while almost entirely ignoring the reinforcement learning literature, arguing the RL is too expensive. This makes the results look much stronger as the authors do not compare against either humans or non-LLM computational agents. As only winrates against other LLM agents and random mover/minmax are presented it's not possible to evaluate the model and judge the presented results. Comparing to something like pluribus would make it clear just how low performing this is.

Looking at the paper as one related to LLMs the authors do not expand our understanding of LLMs significantly. LLMs being bad at tasks requiring strong grounding is not a surprise and using tree search to slightly improve it is also not surprising. The paper also presents no theory or other results that suggest their results will generalize, which limits the usefulness of their results significantly.

I also found the paper to be difficult to follow, the main algorithm is never explicitly stated, instead there's some simple RL like math and a diagram with almost no explanation.

**Questions:**

Figure 3 is very hard to read, can you make the clearer, and use another colour than that bright green.
Was minimax excluded from Tic-Tac-Toe because it has optimal play?
Can you stat in a few sentences the TALAGA algorithm?
Why was there no comparison to non-LLM agents? And why was GPT-3.5 chosen?

---

### Official Review · Reviewer_8Ys6 · 2023-11-09

**Soundness:** 3 good
**Presentation:** 2 fair
**Contribution:** 2 fair
**Rating:** 3
**Confidence:** 4

**Summary:**

This work evaluates the strategic capabilities of gpt-3.5-turbo in two strategy game environments (Tic-Tac-Toe and Texas Hold'em). Through the evaluation, the study finds that the model's performance is only slightly better than that of a random agent, and some reasoning methods are largely ineffective. The work attributes this to the model's inability to think ahead and its severe hallucinations. To address these issues, the authors follow the minimax gaming approach and introduce a recursively-think-ahead mechanism into the model, thereby improving its strategic capabilities.

**Strengths:**

Significance

This paper investigates an interesting question, that is, whether LLMs possess strategic capabilities. Through strategic game experiments, the paper finds that the strategic capabilities of gpt-3.5-turbo is relatively poor. This phenomenon brings new insights to the LLM community, indicating that LLMs, as game agents, still lack strategic and game-theoretic thinking.

---

Originality

This work is application-oriented for LLMs and attempts to apply LLMs to game decision-making, which is not uncommon in related research [1, 2]. The innovation of this paper lies in introducing the conventional optimization-based minimax gaming method into the decision-making of gpt-3.5-turbo, which to some extent improves its strategic capabilities.

---

Quality

The paper conducts a fair comparison of various LLM reasoning methods in two strategic games based on gpt-3.5-turbo. However, the paper lacks discussion on the LLM itself. For example, do other LLMs have the same problem? What kind of LLMs would have this issue? Is the proposed method robust for other LLMs as well? More questions can be found in the questions section.

---

Clarity: The paper provides clear descriptions of the problem and the core ideas of the method. However, as an application-oriented research, some details of the method are missing, making it difficult for other researchers to reproduce the work.

---

[1] Wang, Guanzhi, et al. Voyager: An open-ended embodied agent with large language models. 2023.

[2] Zhu, Xizhou, et al. Ghost in the Minecraft: Generally Capable Agents for Open-World Enviroments via Large Language Models with Text-based Knowledge and Memory. 2023.

**Weaknesses:**

1. This paper aims to apply LLMs to game decision-making and provide strategic capabilities for LLMs from the perspective of reasoning methods. However, there is a lack of discussion on LLMs themselves.
- First, the paper only tests gpt-3.5-turbo, so it is unclear whether this issue is a common problem for LLMs. Do other LLMs, such as GPT-4, Claude-2, and LLaMA2, also have this problem?
- Second, the paper only describes the phenomenon but lacks an in-depth discussion of LLMs. For example, what parameter size of LLMs would lead to this issue? Should LLM training data include relevant game knowledge, such as the capabilities to evaluate game situations? What capabilities should LLMs possess for the reasoning method to be effective? If the method relies on game-related training data, how much data is sufficient?
- Third, how can it be ensured that the proposed method, TALAGA, is robust for all LLMs? The paper lacks descriptions and experiments on the preconditions, i.e., the settings and assumptions for LLMs.

2. The experiments show that the capability of TALAGA is strongly related to the maximum allowed steps to think ahead k, and the number of candidacy actions n. For more complex games, such as Go and Minecraft, k and n need to be significantly increased, which will pose considerable limitations on efficiency and memory.

3. Although the method improves the reasoning ability of LLMs, the win rate against ToT (56%) is still much lower than that of MinMax (81.50%). The effectiveness of this method relies on a large amount of prompt engineering, so what is the advantage of this method compared to guiding LLMs to directly call the MinMax interface?

4. There are some unclear descriptions in the "RECURSIVELY THINK AHEAD" section.
- The role of the Main Actor seems to be only to provide some candidate actions. Can a random agent also be used? If not, does this method strongly rely on the game knowledge embedded in the LLM?
- How is the Anticipation Actor obtained? How can it be ensured that its strategy can block the winning of the Main Actor?

5. As one of the core contributions of the paper, the strategic gaming benchmarks lack prompts for Texas Hold'em. Moreover, there is a lack of descriptions of the conversion methods between states, actions, and natural language. This makes it difficult for other researchers to reproduce and experiment with the work.

**Questions:**

Please refer to the Weaknesses section.